# Total Keratin-18 (M65) as a Potential, Early, Non-Invasive Biomarker of Hepatocyte Injury in Alcohol Intoxicated Adolescents—A Preliminary Study

**DOI:** 10.3390/biom11060911

**Published:** 2021-06-18

**Authors:** Katarzyna Zdanowicz, Witold Olanski, Monika Kowalczuk-Kryston, Anna Bobrus-Chociej, Irena Werpachowska, Dariusz Marek Lebensztejn

**Affiliations:** 1Department of Pediatrics, Gastroenterology, Hepatology, Nutrition and Allergology, Medical University of Bialystok, ul. Waszyngtona 17, 15-274 Bialystok, Poland; monika.kowalczuk-kryston@umb.edu.pl (M.K.-K.); anna.bobrus-chociej@umb.edu.pl (A.B.-C.); irena.werpachowska@udsk.pl (I.W.); lebensztejn@hoga.pl (D.M.L.); 2Department of Pediatric Emergency Medicine, Medical University of Bialystok, ul. Waszyngtona 17, 15-274 Bialystok, Poland; KMRD@umb.edu.pl

**Keywords:** apoptosis, Keratin-18, M65, ethanol, adolescents

## Abstract

Background: Underage drinking is associated with health risk behaviors. Serum keratin-18 (CK18) levels are increased in liver diseases and may be biomarkers of outcome. The purpose of this study was to determine if the total CK18 (M65) or caspase-cleaved CK18 (M30) levels were different in adolescents admitted to hospital because of alcohol intoxication and controls with excluded liver diseases. Methods: A prospective study included 57 adolescents after alcohol use and 23 control subjects. The concentrations of M30 and M65 in the serum samples were evaluated using an enzyme-linked immunosorbent assay. Results: The median age was 15 (14–17) years and 49% were male. There were significant differences in M65 levels between the study and control groups (*p* = 0.03). The concentrations of M30 and M65 were insignificant in adolescents divided into subgroups according to blood alcohol concentrations (BAC). Significant positive correlations were found between BAC and M65 levels (*p* = 0.038; r = 0.3). In receiver operating characteristic (ROC) analysis M65 (cut-off = 125.966 IU/l, Se = 70.2%, Sp = 43.5%) allowed to differentiate between patients with and without alcohol intoxication (AUC = 0.66, *p* = 0.03). Conclusion: M65 appears to be a promising non-invasive biomarker of hepatocyte injury during alcohol intoxication in adolescents. Moreover, a higher concentration of M65 may indicate early organ injury before the increase in the activity of liver enzymes, alanine aminotransferase (ALT) and aspartate aminotransferase (AST).

## 1. Introduction

Alcohol use typically begins in adolescence [1]. Data from a World Health Organization (WHO) analysis showed that more than a quarter (26.5%) of all 15–19-year-olds currently drink alcohol, representing 155 million adolescents worldwide. A survey of alcohol consumption prevalence indicates that binge drinking is the highest among 15–19 year olds in the WHO European Region (43.8%), followed by the Americas (38.2%) and the Western Pacific Region (37.9%). Alcohol use and problem drinking during adolescence reflect the drinking behavior of the general population [2]. Earlier onset of alcohol use is also associated with future problems related to alcohol abuse [3].

Ethanol consumption plays a leading role in adolescent death, including motor vehicle accidents, homicides and suicides [4]. Recent studies show the negative impact of alcohol on the developing brain through changes in verbal learning, visual-spatial processing, memory and attention disorders, as well as deficits in the development and integrity of gray and white matter of the central nervous system [5]. Adolescents with alcohol use disorders appear to be at risk of future behavioral, emotional, social and academic problems.

Non-alcoholic fatty liver disease (NAFLD) is the most common cause of chronic liver disorder and currently affects 10–20% of children and adolescents [6]. However, the number of young patients with alcoholic liver disease (ALD) has been constantly growing, ranging 2.3% between 1988–1994 and 5.1% between 2007–2012. Obesity and alcohol consumption are the fastest growing causes of liver diseases in young people. Alcohol intake can lead to a wide spectrum of liver changes, such as simple steatosis, alcoholic hepatitis (AH), fibrosis, cirrhosis, hepatocellular carcinoma, and accelerates the progression of chronic liver disease (CLD) of an alternative etiology [7]. Drinking alcohol during adolescence predisposes to severe liver disease and its rapid growth is expected in the near future [8]. 

Keratin-18 (CK18), the major intermediate filament protein in hepatocytes, is cleaved by activated caspases during liver cells apoptosis [9]. Caspase-mediated cleavage of cytoskeletal components allows reorganization of the filament network, thereby permitting retraction of the affected cells and membrane blebbing that are noted relatively early in the process of apoptosis [10]. The apoptotic cleavage of CK18 exposes two epitopes, M30 a marker of apoptosis) and M65 (a marker of overall cell death), which can be detected using an enzyme-linked immunosorbent assay (ELISA) [11]. Circulating levels of M30 and M65 have been evaluated as good biomarkers for various liver diseases in adult and pediatric patients. Very few studies have investigated CK18 fragments in ALD in adults. However, no information is available on the association between M30 and M65 and acute alcohol intoxication in adolescents. 

Therefore, the aim of this study was to evaluate the use of serum CK18 fragment levels as a non-invasive marker in detecting liver injury during alcohol exposure in pediatric patients and to more accurately determine the effect of alcohol on liver cell damage.

## 2. Materials and Methods

The protocol was approved by the Bioethics Committee of the Medical University of Bialystok prior to patient recruitment, and the study is in accordance with the Helsinki Accords (approval number: R-I-002/337/2016). This prospective study involved a group of 57 consecutive adolescents (28 boys and 29 girls) aged 14–17 years (median 15 years old) admitted to the Emergency Department (ED) from February 2017 to June 2018 due to suspected acute alcohol intoxication, which was confirmed by measuring the alcohol content in the blood. Informed consent was obtained from all the patients’ parents. None of the subjects had any clinical or laboratory signs of liver diseases and they were not receiving any medications or supplements. Exclusion criteria comprised a history of hepatic virus infections. Patients were also excluded if, when they presented to the ED, they were not medically stable and could not provide consent. The control group consisted of 23 adolescents of similar age and sex who were hospitalized for functional disorders of the gastrointestinal tract.

For all the patients in our study, a blood sample was taken at the time of admission. The blood samples were by immediately centrifuged and frozen at −80 °C for further analysis. Serum samples were collected in tubes containing a silica clot activator. All samples were centrifuged at 1305× *g* for 15 min. Routine blood tests and BAC were performed using standard methods. Headspace gas chromatography/flame ionization detection was used to determine BAC [12]. Serum M65 and M30 levels were quantitatively measured using the M65 EpiDeath ELISA and the M30 Apoptosense ELISA kits (Peviva, Bromma, Sweden). All assays were conducted according to the manufacturer’s instructions. 

Statistical analysis was performed using the Statistica 13.3 package (TIBCO Software Inc., Palo Alto, CA, USA). Data were expressed as median and 25–75 quartiles (Q1–Q3). Because most samples were not normally distributed, the Mann–Whitney U test was applied to test for statistically significant differences between the two groups. Correlations were assessed by Pearson’s correlation test. The receiver operating characteristics (ROC) analysis was performed using IBM SPSS Statistics 20.0. Sensitivity, specificity, positive and negative predictive values were calculated for M65 levels and alcohol intoxication. A *p*-value less than 0.05 was considered statistically significant.

## 3. Results

The clinical and laboratory characteristics of children included in the study are summarized in Table 1. Because of the matching criteria, there were no significant differences in terms of age and sex between the study and control groups. Patients from the study group had only significantly higher serum levels of M65 (*p* = 0.03). There were no significant differences in M30 between both groups. 

In the study group, we observed significantly higher activity of alanine aminotransferase (ALT) (*p* = 0.008) and aspartate aminotransferase (AST) (*p* = 0.004) among boys (Table 2). As shown in Table 3, higher blood alcohol concentrations (BAC) (*p* = 0.02) in older patients were observed in the study group divided according to the age of 12–15 years vs. 16–17 years. Table 4 summarizes the comparisons of M30 and M65 concentrations depending on the BAC; no significant differences were observed. 

A significant positive correlation was found only between BAC and M65 concentrations (*p* = 0.038, r = 0.3). Neither marker correlated with ALT and AST. This finding was confirmed in ROC analysis, the ability of M65 (cut-off = 125.966 U/mL, Se = 70.2%, Sp = 43.5%) to differentiate the adolescents with alcohol intoxication was significant (AUC = 0.66, *p* = 0.03). This cut-off point had a positive predictive value (PPV) of 80% (40/50) and a negative predictive value (NPV) of 43% (13/30) in the test cohort. M30 did not allow a useful prediction to be made (Figure 1).

## 4. Discussion

The effects of alcohol on the liver have been thoroughly studied in adults with acute or chronic alcohol consumption. However, there is a paucity of data indicating the effect of alcohol consumption on liver damage in pediatric patients. Alcohol consumption by young people in particular is an increasing public health problem [13]. Our study is the first to demonstrate the concertation of two cell death markers, M30 and M65 in adolescents hospitalized due to alcohol intoxication. The principal finding of our study is that M65 levels are significantly higher in adolescents admitted for acute alcohol intoxication, which may indicate that a single use of alcohol causes damage to liver cells, despite normal ALT or AST values. Thus, M65 appears to be a more sensitive marker of hepatocyte injury than ALT and AST. Clinical studies have shown that many patients with chronic nonalcoholic hepatitis have normal or slightly elevated ALT levels despite significant inflammation fibrosis as assessed by liver biopsy [14]. Increased serum CK18 levels have been observed in fatty liver, alcoholism, chronic viral hepatitis, autoimmune hepatitis, cholestasis, transplantation, and liver cancer in adult patients [15].

Studies assessing the levels of CK18 fragments in liver disease in the pediatric population focused mainly on NAFLD, which confirmed CK18 as a promising noninvasive marker of fibrosis [16,17]. The usefulness of CK18 in the diagnosis of NASH in children has also been confirmed [18]. Bentel et al. showed that CK18 levels were significantly higher in patients with hepatitis C virus (HCV) compared to healthy controls [19]. However, Darweesh et al. found significant differences in serum CK18 levels, which were highest in NAFLD patients compared to HCV patients and healthy controls [20]. The work by Yilmaz et al. showed that M30 levels were higher in patients with chronic hepatitis B (CHB) compared to healthy controls [21]. Liang et al. extended these findings by analyzing serum levels of M30 and M65 in patients with CHB and NAFLD, stating significantly higher levels of M30 in patients with CHB and NAFLD than patients with CHB non-NAFLD. However, the difference in M65 concentrations between both groups was insignificant [22]. Although the results of studies in patients with viral hepatitis differ from our observations, it is worth noting that depending on the cause, other cell death markers are elevated.

As already mentioned, to the best of our knowledge, our study is the only one to assess the level of markers of apoptosis in adolescents after alcohol consumption. Alcohol use tends to increase during adolescence and is connected to binge drinking, which is defined as drinking ≥5 units of alcohol for boys and ≥4 units of alcohol for girls on a single occasion [13]. According to Silins et al. drinking experiences in adolescents are associated with alcohol-related problems in adulthood [23]. Alcohol is mainly metabolized in the liver. Ethanol and its bioactive products like acetaldehyde-acetate, fatty acid ethanol esters, ethanol-protein adducts are responsible for liver damage. Alcoholic liver disease (ALD) is a broad term including simple steatosis, alcohol-related fatty liver disease (AFLD), AH and cirrhosis [24]. The duration of alcohol consumption and the amount of alcohol are major factors in the development of ALD, however, exact values have not been established. Due to the chronicity of the process, ALD has been reported in the adult population. 

Several studies have been conducted in the adult population with ALD. One recent study involving 824 adult patients found serum levels of M30 significantly elevated, which was a strong predictor of alcoholic steatohepatitis on biopsy. Moreover, the authors suggested the possibility of using M30 as a useful marker in qualifying patients for steroid treatment in severe AH [25]. Woolbright et al. analyzed M30 and M60 levels in patients with alcoholic cirrhosis (AC), severe AH and healthy controls. The highest concentration of M65 was observed in cases of AH. However, in contrast to us, M30 levels were also significantly elevated in patients with AC and AH. Moreover, M65 values in non-surviving AH patients were significantly elevated above their surviving counterparts and healthy controls and may be useful in early stage mortality [26]. In another study, also involving AH subjects, severe cases had significantly higher M30 and M65 levels in comparison with alcohol use disorders in patients without liver injury. On the other hand, similarly to us, no significant correlations of M30 and M65 with ALT and AST were observed. According to the authors, both biomarkers are useful in determining patients at risk of dying within 90 days [27]. 

In a study by Mueller et al. no significant differences were found in the levels of M30 and M65 between adult patients with ALD and NAFLD. In addition, both markers were compared with the results of the histopathological examination and it was noted that M65 and M30 correlate mainly with histological signs of liver damage such as ballooning and Mallory-Denk bodies, followed by lobular inflammation, steatosis, and fibrosis. In contrast to our observation, M65 also correlated with transaminase activities, which may be explained by the inclusion of long-term alcohol users in the study. A surprising finding was an increase in M30 and decrease in M65 in ALD patients undergoing alcohol withdrawal, which may be signs of liver injury improvement and hepatocyte regeneration [28]. Similar correlations of M30 levels with serum liver enzyme levels in adult patients with ALD were observed by Schlossberger et al. In contrast to the previous study, the correlation between plasma M30 levels and fibrosis evaluated in histological examinations was poor, and lack of correlation was observed between M30 concentrations and alcoholic-specific histological features [29]. 

In our study, the ability of M65 to identify adolescents with alcohol intoxication was significant. Bissonnette et al. showed the potential usefulness of M30 and M65 to detect patients with severe alcoholic hepatitis. Both markers were increased in patients with AH. The ability of M30 and M65 to differentiate AH patients from those without AH was high, with an estimated AUC = 0.84 [30]. 

This is the first study in adolescents with alcohol intoxication to investigate concentrations of serum M30 and M65. The novelty of these findings is the main strength of our study and may be an introduction to further analyses in this age group. However, our work has several potential limitations. First, the number of patients was relatively too small to draw definite conclusions. Second, we were unable to monitor the patients included in our study. Also, in comparison to the cited studies involving adults, we did not conduct invasive diagnostics. Due to the lack of indications for liver biopsy [31], possible complications of the procedure, we compared markers of liver damage present in blood serum only. Our study only gives preliminary information about M65 as a potential biomarker of alcohol intoxication, which should be investigated in further researches.

## 5. Conclusions

Underage drinking is more likely to develop alcoholism in adulthood and its consequences. Overall, the data suggest that even one-time alcohol consumption by young patients may contribute to hepatocyte damage. M65 seems to be a promising non-invasive biomarker of hepatocyte injury during alcohol intoxication in adolescents. In addition, M65 appears to be a more sensitive marker of hepatocyte injury than ALT and AST, especially in the early stages of apoptosis and a higher concentration of M65 may indicate early organ injury. Also with regard to the regulation of access to alcohol and preventive measures, special markers are needed to follow-up adolescents who got heavily drunk and were subsequently admitted to ED due to alcohol intoxication.

## Figures and Tables

**Figure 1 biomolecules-11-00911-f001:**
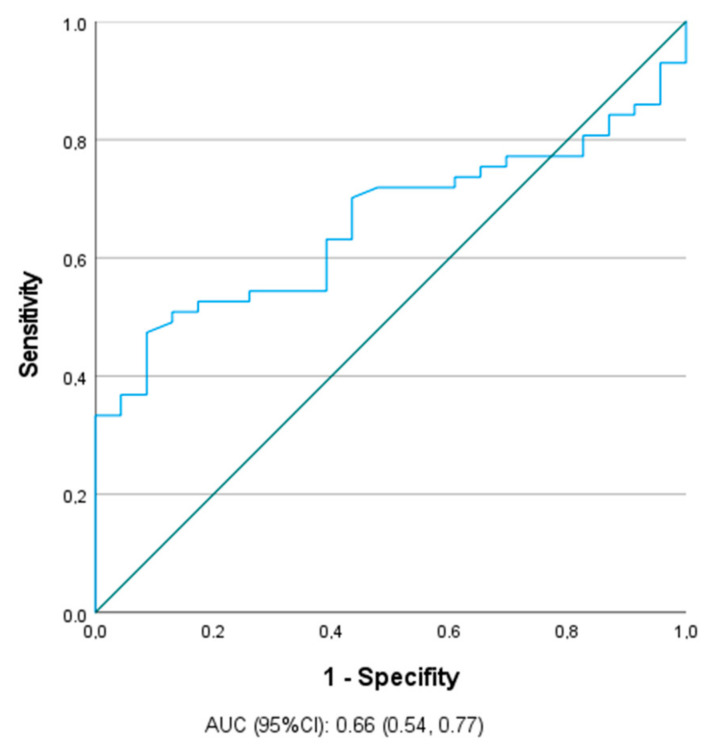
Receiver operating characteristics (ROC) curve of the ability of M65 (cut-off = 125.966 U/mL) to differentiate the adolescents with alcohol intoxication.

**Table 1 biomolecules-11-00911-t001:** Characteristics of the study and control groups: alanine aminotransferase (ALT), aspartate aminotransferase (AST), not significant (NS), not applicable (NA).

Parameter	Study Group (*n* = 57)	Control Group (*n* = 24)	*p*
Alcohol (g/L)	1.78 (1.27–2.22)	-	NA
Age (years)	15 (14–17)	16 (14.5–16.5)	NS
ALT (IU/L)	13 (10–16)	13 (12–14)	NS
AST (IU/L)	22 (18–26)	20 (17–23)	NS
M30 (U/mL)	109.271 (87.988–131.651)	111.981 (95.058–134.055)	NS
M65 (U/mL)	156.878 (114.029–249.763)	120.01 (95.921–145.9)	0.03

**Table 2 biomolecules-11-00911-t002:** Characteristics of the study group in terms of gender: alanine aminotransferase (ALT), aspartate aminotransferase (AST), not significant (NS).

Parameter	Boys (*n* = 28)	Girls (*n* = 29)	*p*
Age (years)	15 (14–17)	16 (15–17)	NS
Alcohol (g/L)	1.825 (1.5–2.18)	1.74 (1.17–2.22)	NS
ALT (IU/L)	14 (12–20.5)	12 (10–14)	0.008
AST (IU/L)	24 (21–28)	19 (16–22)	0.004
M30 (U/mL)	95.058 (85.89–115.5385)	125.014 (87.988–134.989)	NS
M65 (U/mL)	164.886 (119.254–236.394)	140.756 (98.959–258.175)	NS

**Table 3 biomolecules-11-00911-t003:** Characteristics of the study group in terms of age: alanine aminotransferase (ALT), aspartate aminotransferase (AST), not significant (NS).

Parameter	12–15 Years (*n* = 30)	16–17 Years (*n* = 27)	*p*
Alcohol (g/L)	1.64 (1.1–1.96)	2.05 (1.57.227)	0.02
ALT (IU/L)	13 (10.5–15.5)	13 (10–21)	NS
AST (IU/L)	21 (17–27)	22 (19–25)	NS
M30 (U/mL)	98.9595 (85.905–131.651)	109.271 (87.988–133.28)	NS
M65 (U/mL)	142.9615 (75.976–262.372)	164.159 (127.451–248.359)	NS

**Table 4 biomolecules-11-00911-t004:** Characteristics of the study group in terms of BAC: blood alcohol concentrations (BAC), alanine aminotransferase (ALT), aspartate aminotransferase (AST), not significant (NS).

BAC (g/L)	M30 (U/mL)	*p*	M65 (U/mL)	*p*
≤1.15 vs. >1.15 (*n* = 11) vs. (*n* = 46)	90.042 (60.496–130.01) vs. 109.271 (87.988–134.898)	NS	151.032 (75.976–197.331) vs. 162.705 (117.022–267.96)	NS
≤1.50 vs. >1.50 (*n* = 18) vs. (*n* = 39)	127.512 (85.905–136.504) vs. 99.929 (87.988–130.01)	NS	144.421 (89.825–197.331) vs. 164.159 (114.029–267.96)	NS
≤1.78 vs. >1.78 (*n* = 29) vs. (*n* = 28)	109.271 (87.988–131.651) vs. 108.356 (87.988–133.275)	NS	137.809 (124.479–197.331) vs. 180 (106.494–271.446)	NS
≤2.00 vs. >2.00 (*n* = 36) vs. (*n* = 27)	109.271(90.042–131.651) vs. 92.069 (86.905–134.898)	NS	148.1 (126.709–225.08) vs. 164.159 (89.825–267.96)	NS
≤2.20 vs. >2.20 (*n* = 42) vs. (*n* = 21)	109.271 (87.989–131.651) vs. 92.069 (84.8175–132.454)	NS	136.333 (117.022–217.291) vs. 156.878 (114.029–249.763)	NS

## Data Availability

Data supporting reported results are available from the first author to all interested researchers.

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
