# Peer review of "Total Keratin-18 (M65) as a Potential, Early, Non-Invasive Biomarker of Hepatocyte Injury in Alcohol Intoxicated Adolescents—A Preliminary Study"

_biomolecules, 2021, doi:10.3390/biom11060911_

Round 1

Reviewer 1 Report

I am aware of the difficulty to collect AUD cohorts and to perform studies on human samples. I congratulate the 

team to be able to perform the study in this population.

The study is clear and brings a well known non-invasive biomarker to the clinical interpretation. There are negative correlations with transaminases.

However, I would like to see if there is a correlation with gamma GT, since this enzyme is elevated in alcohol consumers. 

Author Response

Thank you for your valuable comments. Due to the inclusion of patients hospitalized in the ED, no gamma GT could be measured. When planning further research, we will try to take this biochemical parameter into account.

Reviewer 2 Report

This study is an interesting investigation. The authors addressed my major concerns in their study limitations statement.

However, the discussion section looks like a literature review. It should focus on the results of the study. The authors need to indicate the sample collection tubes used, what was collected after centrifugation and frozen at -80. For reproducibility, they should indicate the centrifugation speed and time.

Author Response

Thank you for your valuable comments. The article has been supplemented with relevant information in the "Materials and methods" section. The changes made are marked in yellow. Due to the lack of similar studies in the pediatric population, in the discussion we presented the results of our research and included the use of markers of cell death, M30 and M65 in other liver diseases, with particular emphasis on those induced by alcohol consumption.